# Surgery for T4 Colorectal Cancer in Older Patients: Determinants of Outcomes

**DOI:** 10.3390/jpm12091534

**Published:** 2022-09-19

**Authors:** Michael Osseis, William A Nehmeh, Nathalie Rassy, Joseph Derienne, Roger Noun, Chady Salloum, Elie Rassy, Stergios Boussios, Daniel Azoulay

**Affiliations:** 1Department of General Surgery Hôtel-Dieu de France Hospital, Saint-Joseph University, Beirut 1107 2180, Lebanon; 2Department of Hepatobiliary and Liver Transplantation Surgery, Paul Brousse Hospital, Assistance Publique-Hôpitaux de Paris, 75610 Villejuif, France; 3Department of Medical Oncology, Gustave Roussy, 114 Rue Edouard Vaillant, 94800 Villejuif, France; 4Department of Medical Oncology, Medway NHS Foundation Trust, Windmill Road, Gillingham ME7 5NY, UK; 5Faculty of Life Sciences & Medicine, School of Cancer & Pharmaceutical Sciences, King’s College London, London SE1 9RT, UK; 6AELIA Organization, 9th Km Thessaloniki-Thermi, 57001 Thessaloniki, Greece

**Keywords:** postoperative complications, elderly, colorectal cancer, relative survival, T4 tumors

## Abstract

**Background**: This study aimed to compare the outcomes of older and younger patients with T4 colorectal cancer (CRC) treated with surgery. **Methods**: Consecutive patients with T4 CRC treated surgically at Henri Mondor Hospital between 2008 and 2016 were retrospectively analyzed in age subgroups (1) 50–69 years and (2) ≥70 years for overall and relative survival. The multivariable analyses were adjusted for adjusted for age, margin status, lymph node involvement, CEA level, postoperative complications (POC), synchronous metastases, and type of surgery. **Results**: Of 106 patients with T4 CRC, 57 patients (53.8%) were 70 years or older. The baseline characteristics were generally balanced between the two age groups. Older patients underwent adjuvant therapy less commonly (42.9 vs. 57.1%; *p* = 0.006) and had a longer delay between surgery and chemotherapy (median 40 vs. 34 days; *p* < 0.001). A higher trend for POC was reported among the older patients but did not impact the survival outcomes. After adjusting for confounding factors, the overall survival was shorter among the older patients (HR = 3.322, 95% CI 1.49–7.39), but relative survival was not statistically correlated to the age group (HR = 0.873, 95% CI 0.383–1.992). **Conclusions**: Older patients with CRC were more prone to severe POC, but age did not impact the relative survival of patients with T4 colorectal cancer. Older patients should not be denied surgery based on age alone.

## 1. Introduction

Colorectal cancer (CRC) is an age-associated malignancy with nearly 70% of cases diagnosed in individuals older than age 65 and 40% diagnosed in those over 75 years of age [1]. Older patients tend to have a higher prevalence of right colon involvement and mismatch repair-deficient cancers with microsatellite instability, larger and locally invasive CRC, and lower lymph node metastasis [2,3]. The *AJCC Cancer Staging Manual* emphasizes the prognostic role of T4 tumors: T4a if the tumor penetrates the surface of the visceral peritoneum and T4b if the tumor directly invades or is histologically adherent to other organs or structures; the T4 tumors will be staged as IIB (T4aN0M0), IIC (T4bN0M0), IIIB (T4aN1M0), or IIIC (T4bN1M0) [4]. Adjuvant chemotherapy is frequently used in stage III CRC, but it remains controversial for stage II disease. The 5-year disease specific survival for the T4 tumors is about 75.4% [5]; the observed 5-year survival rate for colon cancer in stage IIB (T4aN0M0) was 60.6%, significantly higher than 45.7% for stage IIC (T4bN0M0) [6].

Older patients are commonly under-represented in clinical trials and consequently, the standard therapeutic strategies are not fully validated in this population [7]. Older patients with CRC are difficult to treat for a number of reasons. Aging decreases functional reserve, thus exposing older patients to an increased risk of treatment-related toxicities that are being less than optimally treated. Indeed, patients older than 70 years generally receive 50% less treatment when compared with individuals aged between 35 and 69 years [8]. Moreover, older patients have multiple comorbidities, poor performing status, and late-stage presentations with bowel obstruction and perforations [9,10]. Surgery in patients with T4 CRC is associated with increased postoperative complications (POC) and a morbidity rate of around 30–40% [11]. For these reasons, uncovering the surgical management of this older group—particularly issues specific to POC—will have implications in clinical practice. This paper retrospectively compared disease outcomes and treatment exposures in patients with T4 CRC according to age in order to determine whether differences in age influenced treatment efficacy and toxicity.

## 2. Patients and Methods

### 2.1. Patients and Data Collection

Patients with pathologically proved CRC that were treated at the Department of Surgery, Henri Mondor Hospital were identified in the medical records of patients with CRC treated between January 2008 and December 2016. The study conformed to the principles of the Declaration of Helsinki and was approved by the ethics committee and research board of Henri Mondor Hospital. All patients aged 50 years or above that underwent bowel resection for T4 CRC were selected from this cohort. We retrieved the demographic information, clinicopathologic data, laboratory results, and outcomes data related to each patient from the corresponding medical records of each patient.

### 2.2. Investigations and Treatment Strategies

The management of patients with CRC undergoing elective surgery was discussed during weekly multidisciplinary tumor boards. In an elective setting, patients underwent a diagnostic colonoscopy and a CT scan of the chest, abdomen, and pelvis. Patients with rectal tumors underwent additional pelvic magnetic resonance imaging (MRI) and endorectal ultrasonography to complete local staging. Liver MRI was systematically performed in case liver metastases were suspected. Older patients did not undergo systematic geriatric assessment early in the study period before it became standard practice more recently. Patients with colon cancer underwent partial colectomy with total mesocolon excision. Patients with mid or low rectal cancer received neoadjuvant concomitant chemoradiation (45–50.4 Gy delivered in daily fractions of 1.8–2 Gy over a 5- to 6-week period combined with 5-fluorouracil or capecitabine) followed by total mesorectal excision after 6 to 8 weeks. A shorter neoadjuvant radiotherapy regimen (5 * 5 Gy) was also a possible option followed by total mesorectal excision within 6 weeks. En bloc resection in the case of invasion of adjacent organs and a diverting ileostomy was performed. In patients with liver metastases, hepatic resections were performed simultaneously when feasible using an open or laparoscopic approach [12].

### 2.3. Follow-Up

Patients with locally advanced tumors underwent adjuvant chemotherapy according to the multidisciplinary tumor board recommendations. All patients with T4 rectal tumors operated on the elective bases had neoadjuvant chemoradiotherapy. Patients were then followed up with clinically and radiologically with a CT scan of the chest, abdomen, and pelvis every 3 months for the first 2 years, and then every 6 months for 3 years. A colonoscopy was performed within the first 2 years, and then once every 4 years. MRI and/or positron emission tomography–CT scans were used to rule out disease recurrence and biopsies were done when needed. Patients with metastatic tumors received adjuvant therapy according to the international guidelines and tailored according to tolerability [13,14,15].

### 2.4. Study Outcomes

For the purpose of the study, the age cut-off of 70 years was considered the most appropriate threshold to define the older group [8]. Eligible patients were categorized according to their age at diagnosis into patients aged 50–69 years and those aged ≥70 years. POC was defined by the occurrence of an anastomotic leakage, intraabdominal or pelvic abscess, bleeding, ileus, and wound infection within 90 days after surgery [16]. Anastomotic leakage and grading (A, B, and C) were defined according to the International Study Group of Rectal Cancer [17]. Non-surgical complications included acute kidney injury, pulmonary problems, heart failure, arrhythmias, and all infectious complications. POC were graded according to the Clavien–Dindo grading system and the comprehensive complication index; a novel and more sensitive endpoint for assessing outcome and reducing sample size in randomized controlled trials [16,18]. Severe complications requiring surgical endoscopic or radiologic intervention were graded as III whereas life-threatening complications with organ dysfunction requiring intermediate care were graded as IV. Postoperative mortality was defined by death occurring within the first 90 days after surgery.

### 2.5. Statistical Analysis

Patient demographics and baseline characteristics were expressed as means and standard deviations when normally distributed or as medians and interquartile ranges when non-normally distributed for continuous variables; proportions were used describe categorical variables. Comparisons between groups were performed using Student’s *t*-test or Mann–Whitney U test for quantitative variables and the chi-square test or Fisher test for qualitative variables. Overall survival (OS) was defined by the time elapsed between the date of diagnosis and death or the last follow-up visit. The time interval from surgery to chemotherapy was evaluated to analyze the impact of POCs on chemotherapy administration. Survival curves were obtained with Kaplan–Meier estimates and compared between the two groups with the log-rank test. The relative survival was computed by calculating the ratio of observed to expected survival to adjust survival for life expectancy. The population mortality tables of France delivered by the Human Mortality Database were used to estimate the expected survival (1 January 2021) [19]. The Cox proportional hazard regression was used to identify variables associated with OS; the multivariable analyses were adjusted for adjusted for age, R1 resection, lymph node involvement, CEA level, POC, synchronous metastases, and type of surgery. All *p*-values were two-sided, and the level of significance was set at *p* < 0.05. Data were analyzed using R statistical software (version 3.6.1, R Stats Package, R Foundation for Statistical Computing: Vienna, Austria).

## 3. Results

From January 2008 to December 2016, 115 consecutive patients with pathologically confirmed diagnosis of T4 CRC treated at Henri Mondor Hospital were identified (Figure 1). Of those, nine patients were lost to follow-up, thus 106 patients had complete data and were eligible for analysis. Fifty-seven patients (53.8%) were 70 years or older. The pathological characteristics of the tumors were assessed according to the post-operative findings: 87 patients (82.1%) had colon cancers, 59 patients (55.7%) had lymph node involvement, and 27 patients (25.5%) had synchronous metastases at diagnosis (Table 1).

Overall, the patient baseline characteristics were generally balanced between the two groups except for a higher median carcinoembryonic antigen and cardiovascular comorbidities in the older group (12 vs. 7 µ/L; *p* < 0.001). All patients (80%) operated on electively have had a curative operation. The other 20% operated on in an urgent setting had both curative and palliative surgery (Table 2). Seventy-four patients (69.8%) underwent an open surgery and 32 patients (30.2%) laparoscopically. En bloc resection of adjacent organs was performed in 39 patients (38%) and resection of synchronous liver metastases was performed in 11 patients (10.7%). Twenty-nine patient underwent multivisceral resection. The most involved organ was the uterus with its annexes (posterior exenteration) in eight patients (28.21%). The pathological invasion in the adjacent resected organs was 77% (22/29). R1 resection was reported in eight patients (7.5%). The management plan was similar between the two treatment groups except for lower use of adjuvant therapy (42.9 vs. 57.1%; *p* = 0.006) and longer delay between surgery and chemotherapy (median 40 vs. 34 days; *p* < 0.001) in the older group.

The postoperative complications are summarized in Table 3 according to the age groups. All grade POC and grade III POC occurred in 47 and 12 patients respectively. Six patients (5.7%) had anastomotic leakage: 3 grade A and 3 grade B. Reoperation rate (*n* = 5; 4.7%) was similar between the two groups (Table 3).

After a median follow-up of 33 months, univariable analysis showed that OS was shorter among older patients (HR 1.030, 95% CI 1–1.061), those with R1 margins (HR 3.754, 95% CI 1.273–11.07), and laparoscopic surgery (HR 2.618, 95% CI 1.005–6.803) (Table 4). A multivariable analysis showed that OS was shorter among older patients (HR 3.32, 95% CI 1.491–7.398) (Figure 2) and those with synchronous liver metastasis at diagnosis (HR 2.633, 95% CI 1.102–6.286) (Table 5). Older age as a dichotomized variable was not independently associated with relative survival (HR = 0.873, 95% CI 0.383–1.992) after adjusting survival computation to the expected life expectancy of the study population.

## 4. Discussion

Advances over the last decade have transformed the treatment algorithms of patients with CRC [15,20]; nevertheless, the management of older patients remains complex and is often discussed in multidisciplinary teams [13]. Many patients with CRC with stage II tumors may be managed with surgery alone and those with stage III tumors are at higher risk of relapse and may benefit from adjuvant chemotherapy [13,21,22,23]. T4 CRC constitutes a considerable proportion of stage II and III tumors; the incidence of the T4 CRC is around 5–8.8% and reaches up to 21–43% of advanced resected cases [24,25,26,27,28]. The poor prognosis of T4 CRC may be explained by the local extension toward some structures or organs and the increased risk of lymph node and distant metastases [25]. To our knowledge, this study represents the largest single-center investigation of outcomes based on real-word population of older patients with T4 CRC. A total of 53.8% in the study were 70 years or older. Although we detected a trend for severe POC rate among the older patients, the occurrence of POC did not seem to affect the outcomes of patients with T4 CRC undergoing tumor resection. The older patients underwent adjuvant therapy less commonly and had longer delays between surgery and chemotherapy, probably because of a different tolerability of adjuvant therapy and a potential lower benefit compared with younger patients [29]. Older patients had shorter OS after adjusting for confounding factors; however, the difference in relative survival was not statistically significant after adjustment to the expected life expectancy of the study population. In addition, elderly and younger patients shared the same outcomes in laparoscopic surgery, with equivalent complication rates which supports our idea to unify the treatment approach between those two populations [30,31].

The impact of oncologic surgery among older patients with CRC varied throughout the published literature [9,32,33,34]. The largest series reported discouraging survival outcomes and postoperative morbidities among older patients with CRC. The Colorectal Cancer Collaborative Group has examined the outcomes of surgery among 22,594 elderly and 11,600 young CRC patients treated two decades ago [9]. Compared with patients aged <65 years, older patients had a shorter survival with a 2-year relative survival of 0.91, 0.77, and 0.62 in the 65–74, 75–84, and ≥85-year age groups, respectively. On the other hand, the differences in cancer-specific survival were dismal and the curative intent of surgery decreased significantly among patients in the ≥85-year age group. Older patients present a higher postoperative mortality rate (median 3%, 6.4%, 8.6%, and 19.4% in the age groups below 65, 65–74, 75–84, and ≥85 years, respectively) [9]. A more recent cohort of 895 CRC patients showed that the older patients (31% being 75 years and older) had a higher in-hospital mortality rate (1% vs. 4.2%; *p* = 0.002), shorter survival (5-year OS 68.7% vs. 57.3%; *p* = 0.036), and similar cancer-specific survival [31,33]. In addition, elderly and young patients shared the same outcomes in laparoscopic surgery, with equivalent survival and complications rates.

There are some limitations to be acknowledged in this study, primarily the retrospective nature of the study. Most certainly, there may be a number of older patients that were precluded from surgery due to their poor performance status and comorbidities. These patients were not included in the database of the surgery department and presumably impose a selection bias. Most importantly, we did not have the required information to compute comorbidity or frailty scores—such as the Charlson Comorbidity Index—and the older patients diagnosed early during the study period did not undergo systematic geriatric assessment [35]. Although it is currently common practice to perform a comprehensive geriatric evaluation before surgery which includes the G8 score, many surgeons—especially in underserved areas—do not have access to such evaluations and may omit surgeries among older patients [36,37]. This is probably the main reason that older patients with cancer are commonly undertreated. The study is also limited by the small sample size with comparatively small numbers in the two age groups. The study did not include the nutritional status before surgery, which presents a common variation between young and elderly. Last, we did not have complete data concerning adjuvant therapy details which is considerably less tolerated among older patients and may impact overall survival. When considering the limitations cited above, we definitely assume that our work cannot be considered a generalized result for elderly patients undergoing operation for T4 CRC.

## 5. Conclusions

Older patients with T4 CRC were more prone to severe POC, but age did not impact survival outcomes. For this reason, older patients should not be denied surgery for T4 CRC based on age alone. The prognosis of older patients may be confounded by differences in stage at presentation, tumor site, preexisting comorbidities, and type of treatment received. Older patients should benefit from comprehensive geriatric and preoperative risk assessment outside of urgent surgical indications.

## Figures and Tables

**Figure 1 jpm-12-01534-f001:**
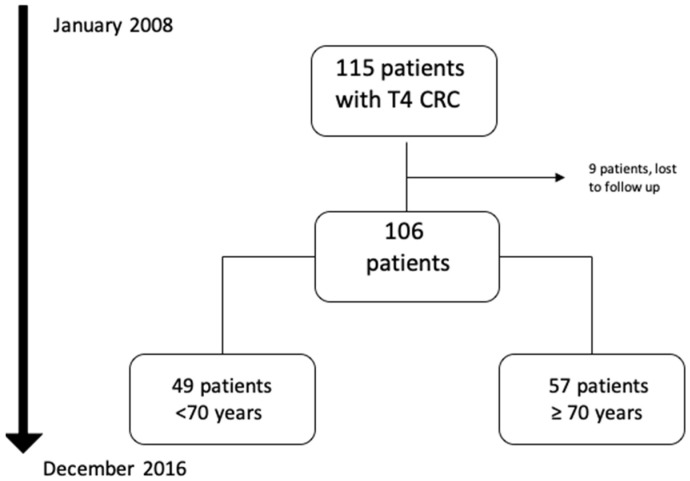
Flowchart of included patients.

**Figure 2 jpm-12-01534-f002:**
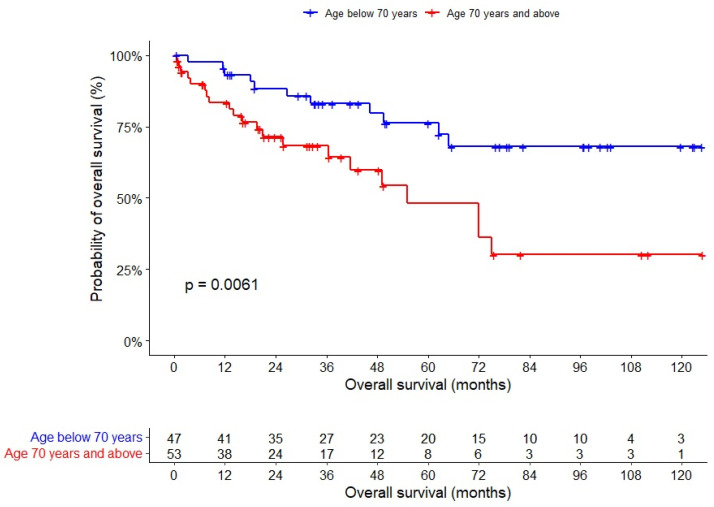
Overall survival of patients with T4 colorectal cancers.

**Table 1 jpm-12-01534-t001:** Patients and tumor characteristics.

Baseline Characteristics		Total *n* = 106	Patients Aged 50–69 Years N = 49 (46.2%)	Patients Aged ≥ 70 Years N = 57 (53.8%)	*p*-Value
Age (years)	Median	71.5	58	80	<0.001
	IQR	21	9	11	
Gender	Male	44 (41.5)	30 (28.3)	14 (13.2)	<0.001
	Female	62 (58.5)	19 (17.9)	43 (40.5)	
American Society of Anaesthesiologists score	<2	78 (73.6)	39 (36.7)	39 (36.7)	0.193
	≥3	28 (26.4)	10 (9.4)	18 (16.9)	
Comorbidities	Cardiovascular	42 (39.6)	14 (13.2)	28 (26.4)	0.031
	Pulmonary	16 (15.1)	6 (5.6)	10 (9.4)	0.447
	Diabetes	19 (17.9)	7 (6.6)	12 (11.3)	0.365
Localization	Rectum	19 (17.9)	10 (9.4)	9 (8.4)	0.536
	Colon	87 (82.1)	39 (36.7)	48 (45.2)	
Specified localization	Rectum	19 (17.9)	10 (9.4)	9 (8.4)	0.296
	Right colon	39 (36.4)	17 (16)	22 (20.7)	
	Transverse colon	7(6.2)	3 (2.8)	4 (3.7)	
	Left colon	42(39.5)	21(19.8)	21(19.8)	
Lymph node	N+	57(53.7)	25(23.5)	32(30.1)	0.389
	N−	46(46.3)	25(23.5)	21(19.8)	
Synchronous metastasis	Stage IVA (liver only)	18 (16.9)	10 (9.4)	8 (7.5)	0.383
	Stage IVA (lung only)	4 (3.7)	3 (2.8)	1 (0.9)	0.334
	Stage IVB	5 (4.7)	4 (3.7)	1 (0.9)	0.179
Serum carcinoembryonic antigen (µ/L)	Median	8.5	7	12	<0.001
	IQR	29	29	29	

IQR: interquartile range.

**Table 2 jpm-12-01534-t002:** Treatment approach.

Treatment Modality		Total *n* = 106	Patients Aged 50–69 Years N = 49 (46.2%)	Patients Aged ≥ 70 Years N = 57 (53.8%)	*p*-Value
Neoadjuvant radiotherapy or chemotherapy		17 (16.1)	9 (8.5)	8 (7.5)	0.546
Operative setting	Elective surgery	85 (80.2)	40 (37.7)	45 (42.4)	0.729
	Urgent surgery	21 (19.8)	9 (8.5)	12 (11.3)	
Surgical procedure	Segmental resection	81 (76.4)	37 (34.9)	44 (41.5)	0.836
	Anterior resection	13 (12.3)	6 (5.6)	7 (6.6)	
	Hartmann’s procedure	6 (5.7)	2 (1.9)	4 (3.8)	
	Abdominoperineal resection	2 (1.9)	1 (0.9)	1 (0.9)	
Surgical approach	Open surgery	74 (69.8)	33 (31.1)	41 (38.6)	0.608
	Laparoscopic	32 (30.2)	16 (15)	16 (15)	
Associated resection	None	67 (63.2)	30 (28.3)	37 (34.9)	0.679
	1 organ	23 (21.7)	10 (9.4)	13 (12.2)	
	≥2 organs	16 (15.1)	9 (8.4)	7 (6.6)	
Synchronous liver resection		11 (10.4)	8 (7.5)	3 (2.8)	0.063
Stoma		22 (21.7)	10 (9.4)	12 (11.3)	0.969
Lymph node involvement		59 (55.7)	25 (23.5)	34 (32)	0.372
Surgical margins status	R0	98 (92.5)	46 (43.3)	52 (49)	0.723
	R1	8 (7.5)	3 (2.8)	5 (4.7)	
Adjuvant chemotherapy		63 (59.4)	36 (33.9)	27 (25.4)	0.006
Delay from surgery to chemotherapy, days	Mean	37	34	40	<0.001

**Table 3 jpm-12-01534-t003:** Details of postoperative complications by age groups.

Complications		Patients Aged 50–69 Years *n* = 49 (46.2%)	Patients Aged ≥70 Years *n* = 57 (53.8%)	*p*-Value
Clavien–Dindo grade III–IV		3 (2.8)	9 (8.5)	0.117
Comprehensive complication index	Median	8.7	0	0.697
	IQR	20.9	8.7	
Complications/patient ≥ 1		25 (23.5)	22 (20.7)	0.199
Anastomotic leakage		1 (0.9)	5 (4.7)	0.213
Other infectious complications	Pelvic abscess	1 (0.9)	0 (0)	0.462
	Intra-abdominal abscess	2 (1.9)	4 (3.8)	0.684
	Urinary infection	1 (0.9)	2 (1.9)	1.000
	Wound infection	4 (3.8)	6 (5.7)	0.749
Non-infectious complications	Ileus	7 (6.6)	11 (10.3)	0.476
	Pulmonary failure/pleuresia	2 (1.9)	1 (0.9)	0.594
	Intra-abdominal bleeding	1 (0.9)	0 (0)	0.430
	Cardiac complications	1 (0.9)	1 (0.9)	1.000

IQR: interquartile range.

**Table 4 jpm-12-01534-t004:** Univariate analysis of baseline characteristics and management plan for overall survival in patients with T4 colorectal cancer.

Variable	HR [95% CI]	*p*-Value
**Age more than 70 years** ** *(vs. less than 70 years)* **	1.030 [1–1.061]	**0.048**
**Male sex** ** *(vs. Female)* **	0.664 [0.324–1.361]	0.264
**BMI ≥ 30 kg/m^2^** ** *(vs. < 30)* **	0.756 [0.230–2.483]	0.645
**ASA ≥ 2** ** *(vs. < 2)* **	0.632 [0.241–1.658]	0.351
**Elevated CEA** ** *(vs. low CEA)* **	1.16 [0.558–2.411]	0.690
**Colon** ** *(vs. rectum)* **	0.921 [0.379–2.239]	0.856
**Synchronous liver metastases** ** *(vs. no synchronous metastases)* **	0.839 [0.255–2.757]	0.008
**Neoadjuvant treatment** ** *(vs. no neoadjuvant treatment)* **	0.964 [0.338–2.752]	0.945
**Emergent surgery** **(*vs. elective*)**	1.422 [0638–3.168]	0.389
**Laparoscopic approach** **(*vs. open*)**	0.382 [0.147–0.995]	**0.049**
**Multiple organ resection** **(*vs. no resection*)**	1.074 [0.524–2.2]	0.845
**Synchronous liver resection** ** *(vs. no synchronous liver resection)* **	0.839 [0.255–2.757]	0.772
**N+ status** ** *(vs. N0 status)* **	1.517 [0.740–3.108]	0.255
**R1 margins** ** *(vs. R0 margins)* **	3.754 [1.273–11.07]	**0.017**
**Postoperative complications** ** *(vs. no postoperative complications)* **	1.393 [0.692–2.802]	0.353
**Grade III–IV complications** ** *(vs. no major complications)* **	0.964 [0.294–3.166]	0.952
**CCI Score**	0.990 [0.9631.018]	0.476
**Adjuvant chemotherapy** ** *(vs. no adjuvant chemotherapy)* **	0.936 [0.431–2.031]	0.867

ASA: American Score of Anesthesiologists; BMI: body mass index; CCI: Comprehensive Complication Index; CEA, carcinoembryonic antigen; CI: confidence interval; HR: hazard ratio; OS: overall survival.

**Table 5 jpm-12-01534-t005:** Multivariate analysis of baseline characteristics and management plan for overall survival in patients with T4 colorectal cancer.

Variable	HR [95% CI]	*p*-Value
**Age more than 70 years** ** *(vs. less than 70 years)* **	3.322 [1.491–7.398]	0.003
**Synchronous liver metastasis**	2.633 [1.102–6.286]	0.004
**Laparoscopic approach** **(*vs. open*)**	0.506 [0.224–1.115]	0.078
**R1 margins** ** *(vs. R0 margins)* **	3.043 [0.964–9.603]	0.058

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
