# Peer review of "Surgery for T4 Colorectal Cancer in Older Patients: Determinants of Outcomes"

_jpm, 2022, doi:10.3390/jpm12091534_

Round 1

Reviewer 1 Report

The study aimed to analysed that the outcomes of older and younger patients with T4 colorectal cancer (CRC) treated with surgery. I have some problems.

1. In the table 1, the information of patients don't include N stage, tumor detail location, nutrition and so o. As we known, these aspects affect the occurrence of complications.

2. Patients' information don't contain factors which are important to patients' survival.

Author Response

We thank the reviewers for their valuable comments on our article. Please find below our response to these comments. Corrections are added to the main text in red color.

------------------------------------------------------------------------------------------------------

Reviewer 1:

  1. In the table 1, the information of patients don't include N stage, tumor detail location, nutrition and so o. As we known, these aspects affect the occurrence of complications.

Thank you for your comment. The “N status” and “specified localization” was added in table one. Unfortunately, we did not have in our data information about the nutritional status. This will be added to our study limitations. (line 244-246)

Baseline characteristics

Total

n = 106

Patients aged 50-69 years

N = 49 (46.2%)

Patients aged ≥ 70 years

N = 57 (53.8%)

p-value

Age (years)

Median

71.5

58

80

< 0.001

IQR

21

9

11

Gender

Male

44 (41.5)

30 (28.3)

14 (13.2)

< 0.001

Female

62 (58.5)

19 (17.9)

43 (40.5)

American Society of Anaesthesiologists score

< 2

78 (73.6)

39 (36.7)

39 (36.7)

0.193

≥ 3

28 (26.4)

10 (9.4)

18 (16.9)

Comorbidities

Cardiovascular

42 (39.6)

14 (13.2)

28 (26.4)

0.031

Pulmonary

16 (15.1)

6 (5.6)

10 (9.4)

0.447

Diabetes

19 (17.9)

7 (6.6)

12 (11.3)

0.365

Localization

Rectum

19 (17.9)

10 (9.4)

9 (8.4)

0.536

Colon

87 (82.1)

39 (36.7)

48 (45.2)

Specified Localization

Rectum

19 (17.9)

10 (9.4)

9 (8.4)

0.296

Right colon

39 (36.4)

17 (16)

22 (20.7)

Transverse colon

7(6.2)          

3 (2.8)

4 (3.7)

Left colon

42(39.5)

21(19.8)

21(19.8)

Lymph node                   

           N+

57(53.7)

25(23.5)         

32(30.1)

0.389

           N-

46(46.3)

25(23.5)

21(19.8)

Synchronous metastasis

Stage IVA (liver only)

18 (16.9)

10 (9.4)

8 (7.5)

0.383

Stage IVA (lung only)

4 (3.7)

3 (2.8)

1 (0.9)

0.334

Stage IVB

5 (4.7)

4 (3.7)

1 (0.9)

0.179

Serum carcinoembryonic antigen (µ/L)

Median

 8.5

7

12

< 0.001

IQR

 29

29

29

IQR: interquartile range.

  1. Patients' information don't contain factors which are important to patients' survival.

Thank you . Information of patient characteristics concerning diabetes status, cardiovascular comorbidities and pulmonary complications were added in the table. Other informations like the G8 score  for elderly were missing ; we highlighted those missing infromation in the limitation section . (line 240)

Reviewer 2 Report

The authors conducted a retrospective single-center study including 106 patients to evaluate short and long-term outcomes in older patients with T4 colorectal cancer. Although it appears that POC were more represented among older patients, age had no impact on the relative survival.

I have some questions and comments mainly on the methodology and results of the study.

1) The authors should clarify whether the patients included underwent surgery only with curative intent (20% of them underwent surgery in urgent setting, others were Stage IV...)

2) Please, add a flowchart of patient inclusion/exclusion with reasons.

3) Please, clarify which patients with mid-low rectal cancer received neoadjuvant treatment.

4) As T4 CRCs were the population study, the authors should provide more information on the involvement of other organs (abdominal wall, bladder etc.) as well as clinicopathological details who could affect outcomes.

5) Tables on uni- and multivariate analyses should be provided in the main text (there is no table in the supplementary material).

6) The authors state in the abstract “A higher trend for POC was reported among the older patients but did not impact the survival outcomes.” What was your outcome measure (Overall POC, Overall CD grade III and IV..)? How have you tested the impact of POC on survival?

7) The author should better explain the contrasting results between KM curves (p=0.0061) and survival computation analysis (HR = 0.873, 95% CI 0.383-1.992) when the covariate “age” was evaluated.

8) In the tables, the percentages should refer to each cohort of study. I read 83.3% of anastomotic leakage in the elderly group (please, replace with 5/57=8.7% as well as the others).

9) Please, expand the discussion section by considering the role of laparoscopy in elderly patient for colon and rectal cancer and taking into account:

- Peltrini R et al. Age and comorbidities do not affect short-term outcomes after laparoscopic rectal cancer resection in elderly patients. A multi-institutional cohort study in 287 patients. Updates Surg. 2021 Apr;73(2):527-537. doi: 10.1007/s13304-021-00990-z.

- Seishima R et al. Is laparoscopic colorectal surgery beneficial for elderly patients? A systematic review and meta-analysis. J Gastrointest Surg. 2015 Apr;19(4):756-65. doi: 10.1007/s11605-015-2748-9.

10) As it is a single-center study with a small sample size, authors should not generalize and specify in the conclusion: In this cohort of patients…

Author Response

Replying to reviewers and connection :

We thank the reviewers for their valuable comments on our article. Please find below our response to these comments. Corrections are added to the main text in red color.

------------------------------------------------------------------------------------------------------

Reviewer 2 :

  • The authors should clarify whether the patients included underwent surgery only with curative intent (20% of them underwent surgery in urgent setting, others were Stage IV...)

Thank you for the clarification, this will be added to our text :All patients (80%) operated electively have had a curative operation. The other 20% operated in an urgent setting had both curative and palliative surgery.  (line 159-161)

  • Please, add a flowchart of patient inclusion/exclusion with reasons

In our study, from January 2008 to December 2016, 115 consective patients with pathologically confirmed diagnosis of T4 CRC treated at Henri Mondor Hospital were identified. Of those, 9 patients were lost to follow-up thus 106 patients had complete data and were eligible for analysis. A flow chart illustrating this will be added to our study.

  • Please, clarify which patients with mid-low rectal cancer received neoadjuvant treatment.

Thank you for the clarification, We will be adding to the text “All patients with T4 rectal tumors operated on the elective bases had neoadjuvant chemoradiotherapy”. (Line 92-93)

  • As T4 CRCs were the population study, the authors should provide more information on the involvement of other organs (abdominal wall, bladder etc.) as well as clinicopathological details who could affect outcomes.

Thank you for this information. We will add “pathological invasion” in our main text 22/29 (77%).  We will add too: “Twenty-nine patient underwent multivisceral resection .The most involved organ was the uterus with its annexes (posterior exenteration) in 8 patients (28.21%). The pathological invasion in the adjacent resected organs was 77% (22/29)”. ( Line 164-166)

  • Tables on uni- and multivariate analyses should be provided in the main text (there is no table in the supplementary material).

Thank you . We will put the table in the main text.

  • The authors state in the abstract “A higher trend for POC was reported among the older patients but did not impact the survival outcomes.” What was your outcome measure (Overall POC, Overall CD grade III and IV..)? How have you tested the impact of POC on survival?

         Thank you , we will make it more clear in our article

  • The author should better explain the contrasting results between KM curves (p=0.0061) and survival computation analysis (HR = 0.873, 95% CI 0.383-1.992) when the covariate “age” was evaluated.

 Thank you. We did the calculation of both overall survival and relative overall survival. The KM curve chart in our article is for overall survival and not the relative overall survival. the title of figure one will be changed. (Line 172). The (HR = 0.873, 95% CI 0.383-1.992) is for relative survival which was calculated by referring to survival at a same age.

  • In the tables, the percentages should refer to each cohort of study. I read 83.3% of anastomotic leakage in the elderly group (please, replace with 5/57=8.7% as well as the others)

Thank you for your comment, we will make the changes in all tables

9) Please, expand the discussion section by considering the role of laparoscopy in elderly patient for colon and rectal cancer and taking into account:

- Peltrini R et al. Age and comorbidities do not affect short-term outcomes after laparoscopic rectal cancer resection in elderly patients. A multi-institutional cohort study in 287 patients. Updates Surg. 2021 Apr;73(2):527-537. doi: 10.1007/s13304-021-00990-z.

- Seishima R et al. Is laparoscopic colorectal surgery beneficial for elderly patients? A systematic review and meta-analysis. J Gastrointest Surg. 2015 Apr;19(4):756-65. doi: 10.1007/s11605-015-2748-9.

Thank you. We will add those  references to the article  Line 210-213

Round 2

Reviewer 1 Report

Authors revised some part of this article. However, I also found some aspect needed to be correct before publishment.

1.    Authors said that “A multivariable analysis showed that OS was shorter among older patients (HR 3.32, 95%CI 1.491-7.398) and those with synchronous liver metastasis at diagnosis (HR 2.633, 95%CI 1.102-6.286) (Figure 2). But in figure 2, it only contained the overall survival of patients with T4 colorectal cancers. It is not consistent.

2.    Authors said that “After adjusting for confound factors, the overall survival was shorter among the older patients (HR = 3.322, 95%CI 1.49-7.39) but relative survival was not statistically correlated to the age group (HR = 0.873, 95% CI 0.383-1.992)”. But it doesn’t show in Table. I think authors should add multivariate analysis results in Table 4.

Author Response

Replying to reviewers and connection:

We thank the reviewers for their valuable comments on our article. Please find below our response to these comments. Corrections are added to the main text in red color.

------------------------------------------------------------------------------------------------------

Reviewer 1:

  1. Authors said that “A multivariable analysis showed that OS was shorter among older patients (HR 3.32, 95%CI 1.491-7.398) and those with synchronous liver metastasis at diagnosis (HR 2.633, 95%CI 1.102-6.286) (Figure 2). But in figure 2, it only contained the overall survival of patients with T4 colorectal cancers. It is not consistent.

We thank you for this comment. Indeed, the figure 2 only represents the OS of younger and older patients and it does not represent the patients with synchronous liver metastasis at diagnosis. The sentence in the main text will be corrected as follow: A multivariable analysis showed that OS was shorter among older patients (HR 3.32, 95%CI 1.491-7.398) (Figure 2) and those with synchronous liver metastasis at diagnosis (HR 2.633, 95%CI 1.102-6.286). Line (182)

  1. Authors said that “After adjusting for confound factors, the overall survival was shorter among the older patients (HR = 3.322, 95%CI 1.49-7.39) but relative survival was not statistically correlated to the age group (HR = 0.873, 95% CI 0.383-1.992)”. But it doesn’t show in Table. I think authors should add multivariate analysis results in Table 4.

Thank you. A new table will be added for the multivariate analysis (Table 5).

Reviewer 2 Report

The revised version of the manuscript still has some concerns that need to be addressed:

1) Point 10 of the previous review has not been addressed: “As it is a single-centre study with a small sample size, authors should not generalize and specify in the conclusion: In this cohort of patients…”

2) Please, specify in Fig.1 (flowchart) that 115 T4 CRC patients were identified.

3) In the Results section, multivariable analysis was indicated as Fig.2. However, Fig.2 corresponds to KM with log rank test and there is not a table showing HR values.

4) Could you graphically report relative survival?

Author Response

Replying to reviewers and connection:

We thank the reviewers for their valuable comments on our article. Please find below our response to these comments. Corrections are added to the main text in red color.

------------------------------------------------------------------------------------------------------

Reviewer 2:

1) Point 10 of the previous review has not been addressed: “As it is a single-centre study with a small sample size, authors should not generalize and specify in the conclusion: In this cohort of patients…”

We thank you for your comment. The reviewer is totally right, our results cannot be generalized to a wider population since it is a single institution and a retrospective study. This is added to our manuscript discussion: When considering the limitation cited above we definitely assume that our work cannot be considered a generalized result for elderly patients operated for T4 CRC. Line 252-254.

2) Please, specify in Fig.1 (flowchart) that 115 T4 CRC patients were identified.

Thank you. 115 patients with T4 CRC was added to the flowchart.

3) In the Results section, multivariable analysis was indicated as Fig.2. However, Fig.2 corresponds to KM with log rank test and there is not a table showing HR values.

We thank you for your comment. We have added the table 4 and table 5 that shows the HR in univariate and multivariate analysis. 

4) Could you graphically report relative survival?

Thank you for reporting this but presenting the relative survival with an acceptable curve will be statistically very difficult in our case. We know that representing it with a figure will be  much better and easier to understand.